# Probiotic Properties of Chicken-Derived Highly Adherent Lactic Acid Bacteria and Inhibition of Enteropathogenic Bacteria in Caco-2 Cells

**DOI:** 10.3390/microorganisms10122515

**Published:** 2022-12-19

**Authors:** Pudi Wang, Songbiao Chen, Chengshui Liao, Yanyan Jia, Jing Li, Ke Shang, Jian Chen, Pinghua Cao, Wang Li, Yuanxiao Li, Zuhua Yu, Ke Ding

**Affiliations:** 1Luoyang Key Laboratory of Live Carrier Biomaterial and Animal Disease Prevention and Control, Luoyang 471003, China; 2Laboratory of Functional Microbiology and Animal Health, Henan University of Science and Technology, Luoyang 471003, China

**Keywords:** chicken-derived, lactic acid bacteria, probiotic properties, adhesion, Caco-2 cell

## Abstract

Lactic acid bacteria (LAB) as probiotic candidates have various beneficial functions, such as regulating gut microbiota, inhibiting intestinal pathogens, and improving gut immunity. The colonization of the intestine is a prerequisite for probiotic function. Therefore, it is necessary to screen the highly adherent LAB. In this study, the cell surface properties, such as hydrophobicity, auto-aggregation, co-aggregation, and adhesion abilities of the six chicken-derived LAB to Caco-2 cells were investigated. All six strains showed different hydrophobicity (21.18–95.27%), auto-aggregation (13.61–30.17%), co-aggregation with *Escherichia coli* ATCC 25922 (10.23–36.23%), and *Salmonella enterica* subsp. *enterica* serovar Typhimurium ATCC 13311 (11.71–39.35%), and adhesion to Caco-2 cells (8.57–26.37%). *Pediococcus pentosaceus* 2–5 and *Lactobacillus reuteri* L-3 were identified as the strains with strong adhesion abilities (26.37% and 21.57%, respectively). Moreover, these strains could survive in a gastric acid environment at pH 2, 3, and 4 for 3 h and in a bile salt environment at 0.1%, 0.2%, and 0.3% (*w*/*v*) concentration for 6 h. Furthermore, the cell-free supernatant of *P. pentosaceus* 2–5 and *L. reuteri* L-3 inhibited the growth of enteropathogenic bacteria and the strains inhibited the adhesion of these pathogens to Caco-2 cells. In this study, these results suggested that *P. pentosaceus* 2–5 and *L. reuteri* L-3, isolated from chicken intestines might be good probiotic candidates to be used as feed additives or delivery vehicles of biologically active substances.

## 1. Introduction

Lactic acid bacteria (LAB), a potential probiotic, can produce lactic acid by fermenting sugar. It has a variety of probiotic functions, such as regulating the composition of the gut microbiota, improving the regulation of immunity, and maintaining host intestinal homeostasis [1,2,3]. An important function of LAB is balancing the gut microbiota [4]. LAB can attach to the gastrointestinal tract (GIT), occupy the adhesion receptor of pathogenic bacteria by binding to the intestinal epithelium, and inhibit the biofilm formation of some bacterial pathogens; this leads to preventing the invasion and proliferation of pathogenic bacteria, thereby maintaining the balance of the gut microbiota [5,6]. The adhesion and colonization of LAB to intestinal epithelial cells is the first step of its probiotic effects [7]. Therefore, the adhesion ability of LAB to human or animal GIT is the most important feature for evaluating its probiotic function.

Nowadays, *Escherichia coli* and *Salmonella* spp. are important enteropathogenic bacteria, causing infections in livestock worldwide [8,9]; the ability of these bacteria to adhere to intestinal epithelial cells determines their virulence characteristics [10]. Meanwhile, overuse of drugs is causing the occurrence of resistant enteropathogenic bacteria to be more common and is also causing the problem of drug residues in livestock. Due to drug resistance issues, researchers are mainly focused on different alternative control strategies against parasites [11], viruses [12], and bacteria [13]. LAB as probiotic candidates are an appealing and effective approach to controlling various infectious diseases [14,15]. LAB can inhibit the adhesion of pathogens in GIT and produce antimicrobial substances, thereby reducing the possibility of infectious diseases [8,16,17]. Numerous studies have reported addressing this problem using probiotics. Feeding pigs with *Pediococcus pentosaceus* and *Lactobacillus reuteri* probiotics can improve their gut microbial composition, immunity, and intestinal regeneration process [18]. *L. reuteri* and *P. pentosaceus* are widely present in feces, dairy, soil, and animal intestines. *L. reuteri* is a probiotic that has been used extensively in humans and animals [19], while *P. pentosaceus* is an emerging probiotic candidate; however, its practical application as a probiotic remain unsolved due to lack of knowledge regarding its mechanisms, side effects, usage, and dosage [20]. Moreover, in past studies, these LAB were isolated from dairy, feces, or silage and did not have good gastrointestinal tolerance. The strains isolated from chicken intestines had generally good tolerance to acids and bile salts. *Lactobacillus plantarum* S27, isolated from chicken, had good tolerance to bile salts and improved the food intake and weight of chickens in vivo [21]. However, these LAB did not have a strong adhesion capacity [22]. The colonization of the GIT by LAB is a prerequisite for their probiotic function. Therefore, it is essential to screen highly adherent LAB with probiotic functions.

Because LAB are a potential probiotic with food-grade and Generally Recognized As Safe (GRAS) status, many studies have reported LAB being used as delivery vehicles of vaccines or cytokines [23]. Based on the above observations, this study aimed to screen highly adherent LAB with potential probiotic functions from the intestines of chicken, to increase the diversity of excellent candidates for probiotics used as feed additives and, therefore, to control diseases caused by enteropathogenic bacteria or delivery vehicles of biologically active substances, such as vaccines or cytokines.

## 2. Materials and Methods

### 2.1. Microorganism and Growth Conditions

A total of 6 LAB strains, including *P. pentosaceus* 2–5, *Lactobacillus salivarius* 2–9, 2–13, and 3–8, *Enterococcus faecalis* 2–28, and *L. reuteri* L-3 were isolated from chicken intestines and analyzed. The bacterial strains were grown on MRS agar and broth culture medium (AOBOX, Beijing, China) at 37 °C for 18–24 h. Three pathogens (*Escherichia coli* ATCC 25922, *Salmonella enterica* subsp. *enterica* serovar Typhimurium ATCC 13311, *Staphylococcus aureus* ATCC 6538) were used for this research and grown in LB medium (Aobox, Beijing, China) in a shaking incubator at 37 °C and 200 rpm. After incubation, the cultures were centrifuged (4 °C and 6000 rpm for 10 min) and washed twice with pre-cooled phosphate-buffered saline (PBS, pH = 7.2). To select the appropriate dilution fold to adjust the cell concentration to 10^8^ CFU/mL, the strains were gradient-diluted, and plated on agar plates to count viable colonies [24]. The bacterial pellets were suspended in different solvents at a bacteria concentration of 10^8^ CFU/mL, which were used for different assays.

### 2.2. Cell Surface Properties Assay

The cell surface properties of LAB strains (hydrophobicity, auto-aggregation, co-aggregation with pathogens) were determined based on the method described by Reuben et al. [25] with slight modifications. The pellets of LAB and pathogenic bacteria (*E. coli* ATCC 25922 and *S.* Typhimurium ATCC 13311) cells were re-suspended in the PBS to adjust their optical density at 600 nm (OD_600_) to 0.6 ± 0.05 (A_x_).

#### 2.2.1. Hydrophobicity

Briefly, 1 mL xylene was mixed with a 3 mL LAB cell suspension and vortexed for 3 min. Then, a two-phase system was formed by incubating the mixture at 37 °C for 2 h. The water phase was removed, and its OD_600_ was measured (*A*_t_). The Hydrophobicity rate was calculated as given in Equation (1).
Hydrophobicity (%) = [(*A*_x_ − *A*_t_)/*A*_x_] × 100(1)
where *A*_x_ was the initial OD_600_ of LAB and *A*_t_ was OD_600_ of the water phase after the incubation.

#### 2.2.2. Auto-Aggregation

A total of 4 mL LAB cell suspension was vortexed for 20 s and then incubated for 4 h at 37 °C. The OD_600_ of the supernatant after 4 h of incubation was assessed (*A*_t_). The auto-aggregation rate was calculated as given in Equation (2).
Auto-aggregation (%) = [(*A*_x_ − *A_t_*)/*A*_x_] × 100(2)
where *A*_x_ was the initial OD_600_ of LAB and *A*_t_ was OD_600_ after the incubation.

#### 2.2.3. Co-Aggregation with Pathogenic Bacteria

An equal volume of LAB cell suspension (*A*_x_) was mixed with the pathogen cell suspensions (*A*_y_) and vortexed for 30 s followed by incubation at 37 °C for 4 h. The OD_600_ of the supernatant after 4 h of incubation was measured (*A*_x+y_). The co-aggregation rate was calculated as given in Equation (3).
co-aggregation (%) = [1 − 2 * *A*_x+y_/(*A*_x_ + *A*_y_)] × 100(3)
where *A*_x_ and *A*_y_ were initial OD_600_ of LAB and pathogenic bacteria, and *A*_x+y_ was OD_600_ of the mixture after the incubation.

### 2.3. Adhesion Assay to Caco-2 Cells

The adhesion of LAB to Caco-2 cells was assessed as described by Behbahani et al. [26] with slight modifications. The Caco-2 cell line was purchased from the Shanghai Cell Bank of the Chinese Academy of Sciences (Shanghai, China) and grown in a high-glucose Dulbecco’s Modified Eagle’s Medium (DMEM, Solarbio, Beijing, China), containing 10% fetal bovine serum (FBS, Gibco, Shanghai, China). The cells were passaged when they reached a confluence of approximately 80–90%. The culture medium was changed on alternate days.

The Caco-2 cells were cultured (2 × 10^5^ cells/mL) in 24-well tissue culture plates (Biofil, Guangzhou, China). The fully differentiated Caco-2 cells were used for the adhesion essay. The LAB pellet was re-suspended in DMEM without antibiotics to adjust its density to 10^8^ CFU/mL. For the adhesion assay, the wells of the plates were washed three times with PBS (pH = 7.2) to remove dead cells and their metabolites. The LAB suspension was added to the wells and incubated for 2 h. The wells were washed three times with 500 μL of PBS to remove the non-adherent bacteria and metabolic secretions. Then, the cells containing adherent bacteria were lysed with 500 μL of Triton-X (Biofroxx, Einhausen, Germany) solution (0.1% *v*/*v* in PBS). After incubating for 15 min, the solution containing released bacteria was taken, gradient-diluted, and plated on MRS agar. The adhesion rate was calculated as given in Equation (4).
adhesion ability (%) = (*A*_x_/*A_t_*) × 100(4)
where *A*_x_ was the initial bacteria counts and *A*_t_ was the bacterial adhesion counts.

To observe the Caco-2 cells adhesion of these strains, Caco-2 cell was cultured (2 × 10^5^ cells/mL) in 24-well tissue culture plates containing coverslip. The suspension of LAB cells was prepared as mentioned above and added to Caco-2 cells. After incubating for 2 h, each well was washed, then each well was incubated with 4% (*v*/*v*) paraformaldehyde (Biosharp, Hefei, China) for 15 min to fix Caco-2 cells. After air-drying, the cells were stained using Gram staining. The coverslips were dried overnight. A biological microscope (Soptop, Shanghai, China) was used to observe adherence under 1000× magnification.

### 2.4. Acid Tolerance

The acid tolerance ability of the bacterial strains was determined as described by Talib et al. [27] with slight modifications. The activated bacterial suspension was centrifuged at 4 °C (6000 rpm for 10 min). The pellet was washed twice and re-suspended in different MRS media with pH 2, 3, and 4. The bacterial suspensions were incubated at 37 °C for 3 h, and the samples were obtained from this suspension at 0, 1, 2, and 3 h time-points. Then, after gradient-diluting the samples with PBS, a 100 μL mixture was used to spread on the MRS plate. After 24–48 h of incubation at 37 °C, viable colonies were counted.

### 2.5. Bile Salt Tolerance

The bile salt tolerance ability of the bacterial strains was determined as described by Hai et al. [9] with some modifications. The LAB cells collected from an 18 h culture were centrifuged at 4 °C (6000 rpm for 10 min). The pellet was washed twice with pre-cooled PBS (pH = 7.2), re-suspended in MRS broth with 0.1, 0.2, and 0.3% (*w*/*v*) bile salts, and then incubated at 37 °C for 6 h. The samples were obtained from this suspension at 0, 2, 4, and 6 h time-point. Then, after gradient-diluting the samples with PBS, a 100 μL mixture was used to spread on the MRS plate. After 24–48 h of incubation at 37 °C, viable colonies were counted.

### 2.6. Antimicrobial Activity

The antimicrobial activity of the cell-free supernatant (CFS) of LAB was assessed as described by Fonseca et al. [10] with slight modifications. The indicator pathogenic bacteria included Gram-negative bacteria (*E. coli* ATCC 25922 and *S.* Typhimurium ATCC 13311) and Gram-positive bacteria (*S. aureus* ATCC 6538). The LAB was incubated in MRS liquid medium for 24 h at 37 °C, and the CFS of LAB centrifuged at 8000 rpm/min for 5 min was used. The indicator pathogenic bacteria at the concentration of approximately 10^8^ CFU/mL was used and spread on LB agar plates. The wells in the plates (3 wells per plate), which were prepared using Oxford Cup, were filled with 100 µL CFS of LAB and incubated for 30 min at 4 °C to allow the diffusion of CFS into the agar, followed by incubation at 37 °C for 24 h. The bacterial inhibition zones of the LAB were measured.

### 2.7. Anti-Adhesion Assay against Enteropathogenic Bacteria

Three methods (competition, exclusion, and substitution assays) were used to evaluate the anti-adhesion of LAB against enteropathogenic bacteria (*E. coli* ATCC 25922 and *S.* Typhimurium ATCC 13311) on Caco-2 cells as described by Singh et al. [28] with some modifications. The Caco-2 cells were cultured as previously described. The pellets of LAB and enteropathogenic bacteria were re-suspended in DMEM without antibiotics at a concentration of approximately 10^8^ CFU/mL.

In the competition assay, equal volumes of LAB suspensions and enteropathogenic bacteria suspensions were mixed and added to the wells followed by incubation with Caco-2 monolayer cells for 2 h.

In the exclusion assay, the wells were first preincubated with LAB suspensions for 1 h, and then the enteropathogenic bacteria suspension was added to each well. The cell cultures in the presence of bacteria were incubated for an additional 1 h.

In the substitution assay, the wells were first preincubated with enteropathogenic bacteria suspension for 1 h, and then, LAB suspensions were added to each well. The cell cultures in the presence of bacteria were incubated for an additional 1 h.

Subsequently, as described above in Section 2.3, the non-adherent bacteria and metabolic secretions were removed and the adherent bacteria were lysed. The solution was gradient-diluted and spread on LB agar plates. The number of enteropathogenic bacteria colonies was counted. The cells were treated with enteropathogenic bacteria only and were used as a control.

### 2.8. Statistical Analyses

The data were expressed as the mean ± standard deviation (SD) of three independent experiments. Statistical analysis was performed using IBM SPSS statistics 22 (SPSS Inc. Chicago, IL, USA). For inhibition of enteropathogenic bacteria adhesion to Caco-2 Cells of LAB, one-way analysis of variance (ANOVA) was used for the data analysis, and a *p*-value of < 0.05 was considered statistically significant.

## 3. Results

### 3.1. Cell Surface Properties of LAB

#### 3.1.1. Hydrophobicity

The hydrophobicity results of six LAB tested are presented in Figure 1. *L. salivarius* 2–9 exhibited the highest hydrophobicity values (95.27%), while *L. salivarius* 2–13, 3–8, and *L. reuteri* L-3 exhibited moderate hydrophobicity with adherence to xylene of 68.20%, 62.68%, and 61.15%, respectively. Moreover, *P. pentosaceus* 2–5 and *E. faecalis* 2–28 exhibited weak hydrophobicity of 23.22% and 21.18%, respectively.

#### 3.1.2. Auto-Aggregation

The auto-aggregation results of the six strains are demonstrated in Figure 2. Among all the tested strains, *L. salivarius* 2–9 showed the strongest auto-aggregation ability (30.17%). The other five strains showed almost similar auto-aggregation abilities, varying from 13.61% to 17.25%.

#### 3.1.3. Co-Aggregation

The co-aggregation results of LAB with pathogenic bacteria (*E. coli* ATCC 25922, *S*. Typhimurium ATCC 13311) are shown in Figure 3. All the strains could co-aggregate with the two pathogens. Almost all the strains had a better co-aggregation ability with *S.* Typhimurium ATCC 13311 as compared to *E. coli* ATCC 25922 except for *L. reuteri* L-3. *L. salivarius* 2–9 showed significant higher co-aggregation ability than *P. pentosaceus* 2–5 or *L. reuteri* L-3.

### 3.2. Adhesion of LAB to Caco-2 Cell Line

Figure 4 shows the adhesion abilities of tested strains to Caco-2 cells. The adhesion rates of all six strains were almost over 10% (Figure 4a). *P. pentosaceus* 2–5 and *L. reuteri* L-3 had the stronger adhesion abilities (26.37% and 21.57%, respectively), followed by *L. salivarius* 2–9 (14.63%), *E. faecalis* 2–28 (12.5%), *L. salivarius* 2–13 (10.37%), and *L. salivarius* 3–8 (8.57%), showing medium adhesion abilities. The adhesion abilities of *L. salivarius* 2–9 showed significantly weaker than *P. pentosaceus* 2–5 and *L. reuteri* L-3. Overall, all the selected six strains showed good adherence ability in vitro. The adherence of strains to Caco-2 cells was observed under a microscope (Figure 4b).

### 3.3. Survival of LAB in an Acidic Environment

The acid tolerance of *P. pentosaceus* 2–5 and *L. reuteri* L-3 was assessed in this study (Figure 5). The two strains demonstrated moderate ability to survive an acidic environment (pH 4.0 and pH 3.0) after 3 h of incubation. In addition, *L. reuteri* L-3 could survive in the acidic environment of pH 2.0 with a survival rate of 48.32% after 3 h of incubation, while that of *P. pentosaceus* 2–5 was less than 1%.

### 3.4. Survival of LAB in Bile Salt Environment

The strains *P. pentosaceus* 2–5 and *L. reuteri* L-3 showed good tolerance in the 0.1% bovine bile salt environment after 6.0 h of incubation with a survival rate of more than 75% (81.08% and 79.76%, respectively) (Figure 6). Moreover, *P. pentosaceus* 2–5 and *L. reuteri* L-3 also showed moderate tolerance in the 0.2% bovine bile salt environment (28.68% and 32.64%, respectively) and cell survival rate of them in the concentration of 0.3% bovine bile salts was less than 1% after 6 h of incubation.

### 3.5. Antimicrobial Activity of CFS of LAB

The antibacterial activities of *P. pentosaceus* 2–5 and *L. reuteri* L-3 were evaluated against pathogens (*E. coli* ATCC 25922, *S.* Typhimurium ATCC 13311, and *S. aureus* ATCC 6538) (Figure 7). Both *P. pentosaceus* 2–5 and *L. reuteri* L-3 showed antibacterial activities against the three pathogens with a diameter of inhibition zone greater than 15 mm. *P. pentosaceus* 2–5 and *L. reuteri* L-3 showed the highest inhibition zone against *S. aureus* ATCC 6538 (22.33 mm and 19.33 mm, respectively), followed by *E. coli* ATCC 25922, and *S.* Typhimurium ATCC 13311. *P. pentosaceus* 2–5 showed higher inhibition zones as compared to those of *L. reuteri* L-3 against the three pathogens used.

### 3.6. Anti-Adhesion of LAB to Enteropathogenic Bacteria on Caco-2 Cells

The competence of the strains to inhibit the adhesion of pathogens to the epithelial Caco-2 cells was measured (Figure 8). A significant reduction was observed in the three inhibition assays against the two enteropathogenic bacteria (*E. coli* ATCC 25922 and *S.* Typhimurium ATCC 13311). The ability of *L. reuteri* L-3 to inhibit the adhesion of *E. coli* ATCC 25922 in the three inhibition assays was better than that of *P. pentosaceus* 2–5. On the contrary, against *S.* Typhimurium ATCC 13311, *P. pentosaceus* 2–5 showed a slightly stronger inhibitory ability than that of *L. reuteri* L-3 in competition and exclusion tests, while *L. reuteri* L-3 had a stronger inhibitory ability than that of *P. pentosaceus* 2–5 in the substitution experiment.

## 4. Discussion

LAB, a potential probiotic, have various probiotic functions, such as inhibiting the intestinal colonization of pathogenic bacteria [29], regulating the balancing of the gut microbiota [30], and improving the immunity of the host [31]. LAB are widely used in the field of food [32], medicine [33], and feed [20]. Intestinal colonization is a prerequisite for LAB to perform probiotic functions [34]. Except for their intestinal colonization and adhesion abilities, the specific characteristics of LAB for use as probiotic candidates are essential to assess.

This study first focused on investigating the adherence ability of LAB, which are the most important potential probiotic strains. The adhesion of LAB can be divided into specific and non-specific adhesion. Specific adhesion is the recognition and combination of LAB to intestinal epithelial cell surface receptors; non-specific adhesion refers to the physical and chemical properties of the strain itself, such as the surface composition of the cell wall and the surface hydrophobicity [35]. The specific adhesion abilities of the strains were measured in vitro using the Caco-2 cell lines [36]. Numerous previous studies have reported the adhesion abilities of LAB strains to Caco-2 cells. Damodharan et al. [37] showed that *P. pentosaceus* KID7 had an adhesion potential of 34.7%. El et al. [38] reported that the adhesion potential of *L. reuteri* 1/c24 to Caco-2 cells was 6.43%. In comparison to these studies, the current study showed that *P. pentosaceus* 2–5 and *L. reuteri* L-3 exhibited strong adhesion ability. This also indicated that *P. pentosaceus* 2–5 and *L. reuteri* L-3 had the potential for anti-adhesion activity against enteropathogenic bacteria in the intestine.

The cell surface properties, such as hydrophobicity, auto-aggregation, and co-aggregation, are usually determined to reflect the non-specific adhesion properties of the strains. The hydrophobicity of bacteria determines the non-specific adhesion of bacteria to various biotic and abiotic surfaces and interfaces; auto-aggregation reflects the ability of a bacterial strain to adhere to the intestine; and co-aggregation shows the colonization and infection of pathogenic bacteria in the gut [39,40]. Meanwhile, numerous studies have shown that the adhesion of LAB strains to Caco-2 cells was positively correlated with the cell surface property [41,42]. In this study, the results of the cell surface properties of *P. pentosaceus* 2–5 and *L. reuteri* L-3 showed significantly lower adhesion capability than *L. salivarius* 2–9, and the two strains showed significantly stronger adhesion capability in the cell adhesion assays than *L. salivarius* 2–9. The result was in agreement with that reported by Liu et al. [43]; this study did not observe a positive correlation between the cell surface properties and adhesion of the probiotic strains. These differences in conclusions might be due to the differences in testing methods and specificity of the strains, causing differences in responses between the strains [44]. Based on the results of these experiments, *P. pentosaceus* 2–5 and *L. reuteri* L-3, having high adhesion abilities, were selected for the subsequent experiments.

In addition to the high adhesion abilities of the strains colonizing GIT, the probiotics should survive the GIT environment to reach and colonize the intestine and show active metabolism [45]. The pH of gastric juice in the stomach after the intake of food is usually between 3 and 5, and the bile salts in the small intestine are typically between 0.03 and 0.3%. Moreover, food usually passes through the stomach into the intestines within 2.0 to 3.0 h, while the intestine emptying duration is about 4.0 to 6.0 h [9,46,47]. Both the strains in the present study showed good tolerance to acidic and bile salt environments. Similar results of LAB were reported by researchers [9,27]. *L. reuteri* L-3 showed strong tolerance capabilities at pH 2 (48.32%), which were better than those of *L. reuteri* ZA33 isolated from the silage mixture analyzed by Wang et al. [48]. Moreover, in this study, both the strains showed less than 1% survival ability in the bile salt concentration of 0.3%, while the number of viable bacteria was still over 10^6^ CFU/mL. Gilliland et al. [49] showed that the 2.5 × 10^6^ CFU/mL cells of *Lactobacillus acidophilus* showed beneficial effects in utilizing lactose. The differences in acid and bile salt tolerance of the strains might be due to the origin of the isolates or characteristics of the strains [50], and the wild strains with domestication might possess a better acid or bile salt tolerance [51].

After the successful entrance and colonization of the intestine by probiotics strains, they must have the potential to inhibit the colonization of pathogenic bacteria in the intestine because the key mechanism of intestinal infection is the adhesion of pathogenic bacteria to the mucosal surface of the intestines [52]. The probiotics’ mechanisms of action against enteric pathogens are complex and multifactorial and mainly included the production of inhibitory substances, inhibiting the adhesion of pathogenic bacteria, immune system modulation, and improving barrier function [53]. The inhibitory substances of LAB were organic acids secreted outside the cell. Consequently, the CFS of LAB was usually used for the antimicrobial test in vitro. A previous study showed that the CFS of *Lactobacillus casei* IMAU60214 and *Lactobacillus helveticus* IMAU70129 could significantly inhibit *E. coli*, *S.* Typhimurium, and *S. aureus* growth; these results were similar to those in the present study [44].

The adhesion of intestinal pathogenic bacteria to intestinal epithelial cells is an important step to cause disease. The pathogenic bacteria bind to the adhesion sites of intestinal epithelial cells and form tight junctions, proliferate, and produce enzymes or toxins to cause disease to the host; they also form a dense biofilm of pathogenic bacteria, which increases the harm to host cells [54]. The highly adhesive LAB can inhibit the activity of pathogenic bacteria except for the endocrine organic acids that colonize the intestinal tract, and they can also reduce the interaction between pathogenic bacteria and intestinal mucosa by occupying the adhesion sites of epithelial cells, and inhibit the formation of pathogenic bacteria biofilm [55,56,57]. Since *S. aureus* did not act directly on the intestinal epithelium by its enterotoxins [44], the anti-adhesion abilities of the strains against *E. coli* and *S.* Typhimurium were evaluated. The strains showed a significant reduction in the adhesion of *E. coli* and *S.* Typhimurium. Compared to the results obtained by Vasiee et al. [58] and Singh et al. [28], similar results were observed that the strains were able to antiadhesion against *E. coli* and *S.* Typhimurium by competition and exclusion, while the strains used in this study exhibited inhibition abilities in the substitution experiments.

*P. pentosaceus* 2–5 and *L. reuteri* L-3, which were screened from the intestine of chicken, had beneficial functions, particularly high adherent ability, and could be defined as probiotic candidates by the results of the present study, and both strains were rare in recent studies. They will increase the diversity of excellent candidates for probiotics used as feed additives to improve drug resistance and control of diseases caused by enteropathogenic bacteria or delivery vehicles of biologically active substances, such as vaccines or cytokines. However, in order to use these two strains in clinical applications, it is necessary to conduct further research such as feeding livestock to evaluate the safety and practical application effects of strains, and assess their potential for biotechnology development.

## 5. Conclusions

In this study, the cell surface properties and adhesion abilities of six LAB strains derived from chicken were tested. *P. pentosaceus* 2–5 and *L. reuteri* L-3 showed high adhesion abilities and could survive the gastric acid and bile salt environments. The CFS of *P. pentosaceus* 2–5 and *L. reuteri* L-3 showed inhibitory effects, and the two strains could inhibit the colonization and invasion of enteropathogenic bacteria. In short, *P. pentosaceus* 2–5 and *L. reuteri* L-3 isolated from chicken exhibited high adhesion ability and interesting probiotic properties. Therefore, these bacteria strains might be good probiotic candidates to be used as feed additives to improve drug resistance and control various infectious diseases or delivery vehicles of biologically active substances, such as vaccines or cytokines. In addition, animal model tests and an assessment of their potential for biotechnology development are needed prior to production applications

## Figures and Tables

**Figure 1 microorganisms-10-02515-f001:**
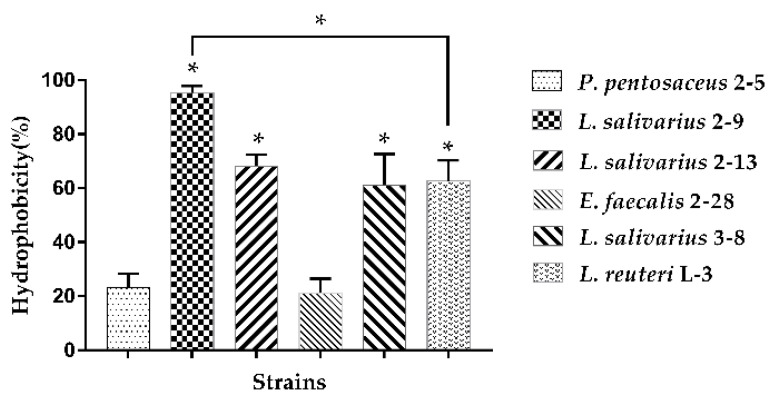
Hydrophobicity of strains determined after 2 h of incubation with xylene. The results represent the mean ± SD of three independent experiments. The significant difference was indicated by asterisks compared to *P. pentosaceus* 2–5: * *p* <0.05.

**Figure 2 microorganisms-10-02515-f002:**
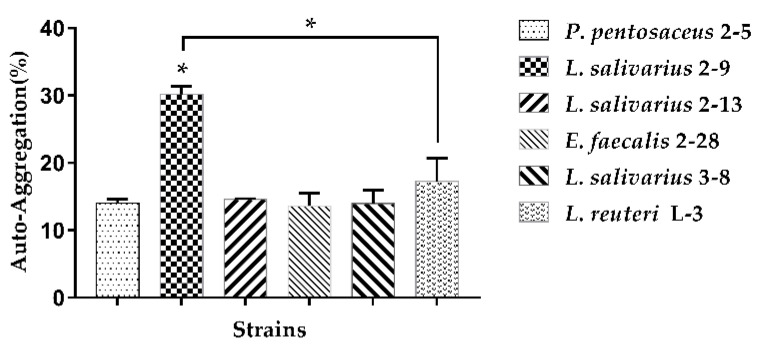
Auto-aggregation of strains determined after 4 h of incubation at 37 °C. The results represent the mean ± SD of three independent experiments. The significant difference was indicated by asterisks compared to *P. pentosaceus* 2–5: * *p* <0.05.

**Figure 3 microorganisms-10-02515-f003:**
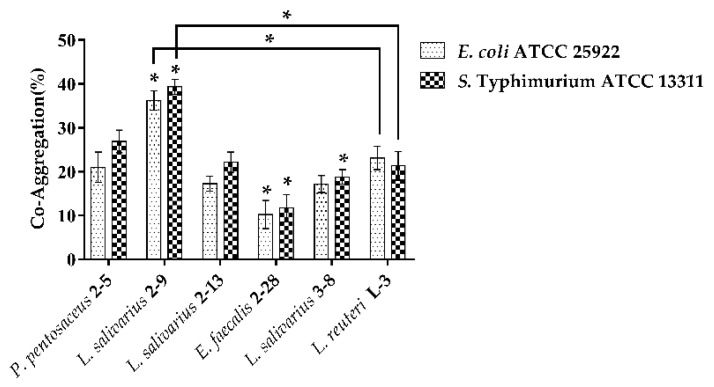
Co-aggregation of strains with pathogenic bacteria was determined after 4 h of incubation at 37 °C. The results represent the mean ± SD of three independent experiments. The significant difference was indicated by asterisks compared to *P. pentosaceus* 2–5: * *p* <0.05.

**Figure 4 microorganisms-10-02515-f004:**
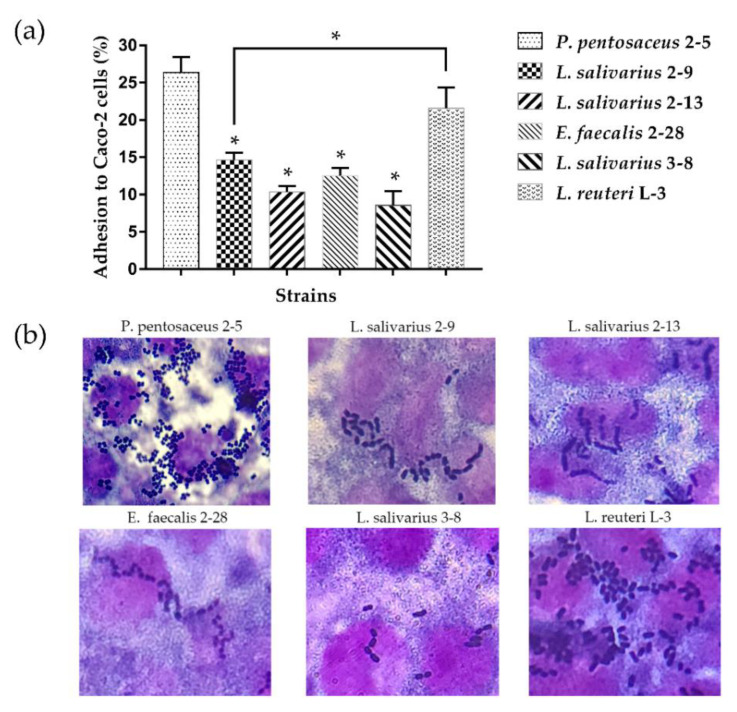
Strains adherence to Caco-2 cells: (**a**) Percentage of strains adherence to Caco-2 cells. (**b**) Representative microscopic observation (1000×). The results represent the mean ± SD of three independent experiments. The significant difference was indicated by asterisks compared to *P. pentosaceus* 2–5: * *p* <0.05.

**Figure 5 microorganisms-10-02515-f005:**
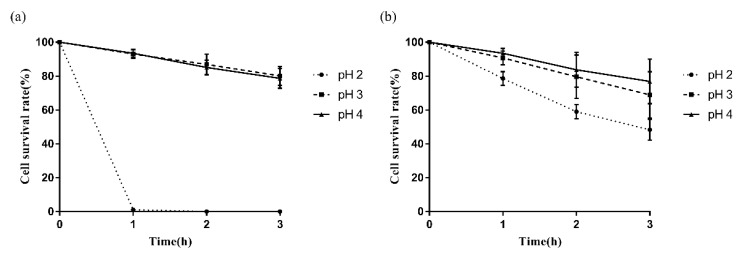
The survival rate of *P. pentosaceus* 2–5 (**a**) and *L. reuteri* L-3 (**b**) in different acidic environments after 1 h, 2 h, and 3 h of incubation at 37 °C. The results represent the mean ± SD of three independent experiments.

**Figure 6 microorganisms-10-02515-f006:**
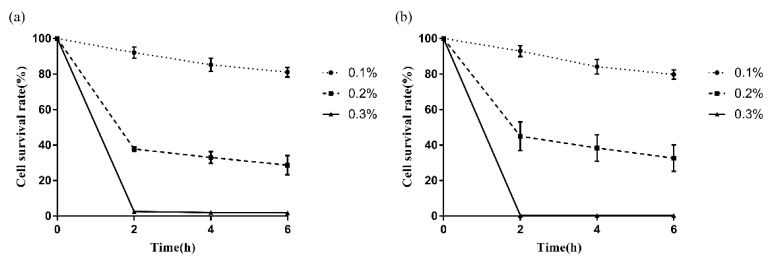
The survival rates of *P. pentosaceus* 2–5 (**a**) and *L. reuteri* L-3 (**b**) in the presence of bile salts after 2 h, 4 h, and 6 h of incubation with different concentrations (% *w*/*v*). The results represent the mean ± SD of three independent experiments.

**Figure 7 microorganisms-10-02515-f007:**
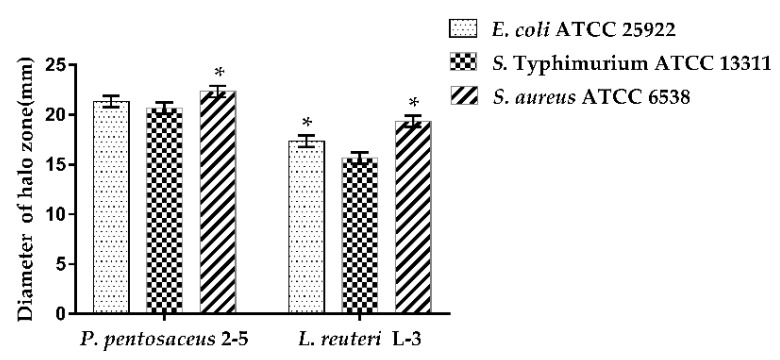
Inhibition zones of the CFS of the strains against pathogenic bacteria. The results represent the mean ± SD of three independent experiments. The significant difference was indicated by asterisks compared to LAB against *S.* Typhimurium ATCC 13311: * *p* <0.05.

**Figure 8 microorganisms-10-02515-f008:**
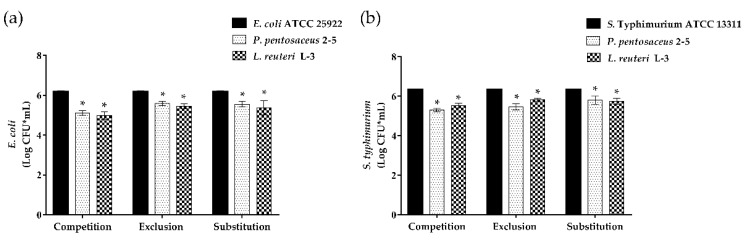
The ability of *P. pentosaceus* 2–5 and *L. reuteri* L-3 to inhibit the adhesion of enteropathogens to Caco-2 cell lines. Caco-2 cells were incubated with *E. coli* ATCC 25922 and *S.* Typhimurium ATCC 13311 alone as a control: (**a**) The ability of strains to inhibit the adhesion of *E. coli* ATCC 25922. (**b**) The ability of strains to inhibit the adhesion of *S.* Typhimurium ATCC 13311. The results represent the mean ± SD of three independent experiments. The significant difference was indicated by asterisks compared to the control: * *p* <0.05.

## Data Availability

Not applicable.

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
