# Peer review of "Probiotic Properties of Chicken-Derived Highly Adherent Lactic Acid Bacteria and Inhibition of Enteropathogenic Bacteria in Caco-2 Cells"

_microorganisms, 2022, doi:10.3390/microorganisms10122515_

Round 1

Reviewer 1 Report (Previous Reviewer 2)

Comments

The objective of this study was to check the probiotic properties of chicken-derived highly Adherent Lactic Acid bacteria and inhibition of enteropathogenic bacteria in Caco-2 cells. Study is still interesting however, below mentioned point should be improved

Line 18: co-aggregation with E. coli ATCC 25922 (10.23–36.23%) “Here” add the comma after (10.23–36.23%)

Line 24-27: These results suggested that P. pentosaceus 2-5 and L. reuteri L-3, isolated from chicken intestines in this study, might be good probiotic candidates to be used as feed additives or delivery vehicles of biologically active substances “modify to” In this study, these results suggested that P. pentosaceus 2-5 and L. reuteri L-3, isolated from chicken intestines might be good probiotics candidates to be used as feed additives or delivery vehicles of biologically active substances      

Line 81-83: To select the appropriate dilution fold to adjust the cell concentration to 108 cfu/mL, the strains were gradient-diluted, and plated on agar plate to count viable colonies “here” add the reference after viable colonies

Line 132-133: where ?x was initial bacteria counts and ?t was adhesion bacterial “modify to” bacterial adhesion counts

Line 135: Cells were prepared as above mentioned “modify to” mentioned above

Line 200-201: L. salivarius 2-9 exhibited the highest values of hydrophobicity “modify to” hydrophobicity values

Line 227-229: Significant differences was indicated asterisks compared to P. pentosaceus 2-5 “here” P. pentosaceus should be italic

Line 239-242: LAB adherence to Caco-2 cells. (a) Percentage of LAB adherence to Caco-2 cells “here” what your mean all LAB comes under probiotics definition? Here mentioned the probiotics (bacteria names) or correct it clearly

Line 255-257: Rephrase the sentence; Moreover, these two strains also showed moderate bile salt resistance to 0.2% bovine bile salt, and cell survival rate of them in the bile salt concentration  of 0.3% bovine bile salt were less than 1% after 6 h of incubation.

Line 293-294:  LAB are widely used in the field of food, medicine “after medicine” used the most recent reference such as http://dx.doi.org/10.29261/pakvetj/2021.009

Line 347-351: Rephrase the sentence: The probiotics’ mechanisms of action against enteric pathogens is complex and multifactorial and mainly included production of inhibitory substances, inhibiting the adhesion of pathogenic bacteria through competition, exclusion, substitution, immune system modulation, and improving barrier function

Comment: In this study authors used LAB except the specific names of LAB? Is all LAB come under the definition of probiotics?

Comment: Check the references following journal format in whole the manuscript

Author Response

Reviewer 2 Report (Previous Reviewer 1)

The manuscript was much more improved.

I enclose the comments as a pdf file.

Round 2

Reviewer 1 Report (Previous Reviewer 2)

Comment: Check the references including Journal names used in references in consistant manner

Author Response

This manuscript is a resubmission of an earlier submission. The following is a list of the peer review reports and author responses from that submission.

Round 1

Reviewer 1 Report

·        First sentence in the abstract grammar and style should be corrected.

·        Lines 12-13 and 31: lactic acid bacteria are not probiotics.

·        Line 18-19: please correct your spelling of S. typhimurium ATCC 1813311 according to microbiological demands.

·        Line 20: what does it mean “the best”? It is not scientific.

·        Correct English language in lines 21-22.

·        In line 26 should be “probiotic” not “probiotics”.

·        The authors should standardize and organize the spelling of microbial names of microorganisms. For example, when they use the name of a microorganism for the first time they should use the entire genus and species name and also the strain symbol. After that, they can use the abbreviation of the genus. Do not use the strain symbols alone because it is confusing.

·        The authors mix up the concept of probiotics with LAB. Not all LAB are probiotic/s, but only selected strains in in vitro and then in vivo studies, according to FAO/WHO. Writing about LAB as probiotics is incorrect. E.g. Enterococcus faecalis or some Streptococcus pyogenes strains can be pathogens and they are LAB. Authors should use the term “probiotic” only for selected strains with in vivo proven probiotic properties. In other cases, authors should write “potentially probiotics” or “probiotic candidates”. It should be corrected in the whole manuscript. Thus, the Introduction must be corrected, it is written at a very high general level and it should sound more scientifically.

·        Line 63: what does  it mean “good probiotic functions”?

·        Add detailed aim of the study.

·        What is the novelty of the study? Emphasise it in the text.

·        Line 75: how it was adjusted?

·        Line 157: authors should indicate that they investigated antimicrobial activity of LAB culture supernatants, not live cells of LAB.

·        Methods should be described in a more logical sequence and order. Methodology is described somewhat chaotically.

·        Line 193 – how many experiments were conducted and in how many repeats?

·        Subtitles in the results should be different that in methods.

·        Figures 1,2,3,5,6,7,8 – add statistical analysis. What strains?

·        Figure 4A - – add statistical analysis.

·        Lines 349-350: Is there any correlation between hydrophobicity, aggregation and adhesion? Is there any statistical analysis? On what basis authors made the conclusion?

·        The discussion is very poor.

·        Line 350: these strains are not probiotic.

The manuscript is basic research, suitable for publication in a journal with a lower impact factor.

Reviewer 2 Report

Comments: The objective of this study was to check the probiotic properties of chicken-derived highly Adherent Lactic Acid bacteria and inhibition of enteropathogenic bacteria in Caco-2 cells. Study is interesting however, below mentioned point should be improved

Line 18: co-aggregation with E. coli ATCC 25922 (10.23–36.23%) “here” add the comma after (10.23–36.23%)

Line 25: Taken together, these results suggested that Pediococcus pentosaceus ”here remove” taken together

Line 45-46: Add a paragraph after drug residues in livestock like “Due to drug resistance issue, researchers are mainly focusing on different alternative control strategies against parasite, virus , and bacteria” “here” after parasite add recent references such as http://dx.doi.org/10.29261/pakvetj/2020.043;  https://doi.org/10.1080/00071668.2020.1759787;  https://doi.org/10.1016/j.ygeno.2021.10.019;  http://dx.doi.org/10.29261/pakvetj/2022.020 and https://doi.org/10.3390/life12030449 as well as add references after virus and bacteria. Probiotics (LAB) are appealing and effective approach in controlling various infectious diseases “here” add reference such as https://doi.org/10.1080/00439339.2021.1883412

Line 46: Therefore, LAB can inhibit the adhesion of pathogens in GIT and produce antimicrobial substances “here” remove therefore

Line 74-75: How you adjust the bacteria concentration to 108 CFU/mL in PBS? Explain here clearly

Line 189-190: The cells were treated with pathogenic bacterial only and was used as control “modify to” The cells were treated with pathogenic bacteria only and was used as a control

Line 282: LAB, a type of probiotic, has various probiotic functions “here” remove the type of

Line 282-285: probiotic functions, such as inhibiting the intestinal colonization of pathogenic bacteria “here” used the most recent reference such as https://doi.org/10.1080/00439339.2021.1883412 and improving the immunity of the host “here” add most recent references such as https://doi.org/10.3390/vaccines10010097 “and” LAB are widely used in the field of food, medicine “add” recent reference like http://dx.doi.org/10.29261/pakvetj/2021.009 after medicine

Line 310-311: the probiotics should survive the GIT environment in order to reach and colonize the intestine “here” remove in order

Line 367-484: Carefully check the references following journal format

Comment: What’s about the novelty of this study? Many studies have already done like recent study